# Could listening to music during pregnancy be protective against postnatal depression and poor wellbeing post birth? Longitudinal associations from a preliminary prospective cohort study

Daisy Fancourt,[1,2] Rosie Perkins[2,3]

¹Department of Behavioural Science and Health, University College London, London, UK
²Faculty of Medicine, Imperial College London, London, UK
³Centre for Performance Science, Royal College of Music, London, UK

**Correspondence to**
Dr Daisy Fancourt;
d.fancourt@ucl.ac.uk

## ABSTRACT

**Objectives** This study explored whether listening to music during pregnancy is longitudinally associated with lower symptoms of postnatal depression and higher well-being in mothers post birth.

**Design** Prospective cohort study.

**Participants** We analysed data from 395 new mothers aged over 18 who provided data in the third trimester of pregnancy and 3 and 6 months later (0–3 and 4–6 months post birth).

**Primary and secondary outcome measures** Postnatal depression was measured using the Edinburgh Postnatal Depression Scale, and well-being was measured using the Short Warwick-Edinburgh Mental Well-being Scale. Our exposure was listening to music and was categorised as 'rarely; a couple of times a week; every day <1 hour; every day 1–2 hours; every day 3–5 hours; every day 5+hrs'. Multivariable linear regression analyses were carried out to explore the effects of listening to music during pregnancy on depression and well-being post birth, adjusted for baseline mental health and potential confounding variables.

**Results** Listening during pregnancy is associated with higher levels of well-being (β=0.40, SE=0.15, 95% CI 0.10 to 0.70) and reduced symptoms of postnatal depression (β=−0.39, SE=0.19, 95% CI −0.76 to −0.03) in the first 3 months post birth. However, effects disappear by 4–6 months post birth. These results appear to be particularly found among women with lower levels of well-being and high levels of depression at baseline.

**Conclusions** Listening to music could be recommended as a way of supporting mental health and well-being in pregnant women, in particular those who demonstrate low well-being or symptoms of postnatal depression.

## INTRODUCTION

Perinatal mental health problems affect around 20% of women at some point during the perinatal period.[1] In terms of conditions characterised by negative symptomology, postnatal depression (PND) is one of the

### Strengths and limitations of this study

► This preliminary prospective cohort study tracked a sample of women across the perinatal period providing data at 12-week intervals.
► The data include a rich set of variables on music listening behaviours among participants.
► We adjusted for all identified confounding variables in our analyses and ran sensitivity analyses to test our assumptions.
► The data are not nationally representative, although there is a clear spread of participants from varying socioeconomic backgrounds as well as variations in the levels of exposure and outcome variables.
► As this is a cohort study and not interventional, it is not possible to confirm causality.

most common problems, and is a debilitating condition with symptoms including fatigue, irritability, insomnia and anhedonia; symptoms which in 25% of affected women last for at least 1 year.[2] Over the last two decades, there has been significant research into the effects of PND on mother and infant as well as attention paid to how it can be prevented or managed.[3 4] However, in terms of conditions relating more to the absence of positive symptomology, such as low hedonic or eudemonic well-being, there has been much less research. The few studies that do exist have found that negative mood, as indicated by the Edinburgh Postnatal Depression Scale (EPDS), has a correlation of just −0.46 with positive experiences of motherhood, as indicated by a principal component analysis of six positive experiences of motherhood.[5] This suggests that, as in the wider population, depression and well-being are separate constructs in the context of perinatal mental

health and a positive perinatal experience is more than simply the inversion of negative mood.[6] Building on this, women even without PND have been found to demonstrate impairments in emotional problems and vitality, suggesting that even in the absence of depression, mothers can have impaired well-being.[7] In light of this, a review has argued that psychological well-being defined as a multidimensional construct should be an integral part of maternity care,[8] and a more recent construct analysis has highlighted the importance not just of identifying PND but also of identifying women with suboptimal perinatal well-being and supporting them to achieve positive psychological functioning.[9]

In seeking to support the perinatal mental health of women, the pregnancy period has been highlighted as critical. Prenatal mental health has repeatedly been shown to be one of the largest predictors of postnatal depression[10–12] and well-being.[9] In particular, the third trimester of pregnancy has been identified as an important transition period involving adaptation to emotional and physical changes, leading to feelings of well-being often less pronounced than in the previous trimesters.[13] Early detection of symptoms of depression and low well-being during pregnancy and prompt intervention is therefore important in reducing adverse consequences.

In light of this, there are a number of interventions that have been developed to try and support mental health in the prenatal period as a way of reducing postnatal mental health problems, in particular focusing on the third trimester as a point of intervention. There have been findings that support the application of cognitive-behavioural and interpersonal psychotherapy, suggesting that depression following childbirth could be prevented by brief interventions in the prenatal period.[14 15] In exploring other interventions, prenatal hypnotherapy has been found to significantly reduce PND and improve psychological well-being at 2 weeks and 10 weeks postpartum.[16] And a psychosocial intervention involving group meetings to discuss aspects of parenthood in the final trimester of pregnancy and first 6 months postpartum has been found to reduce PND among first-time mothers.[14] However, as many mothers continue to work full time until shortly before their due dates, in-person interventions may not be feasible for all mothers and are of course limited by what is available in different geographical areas. As a result, there is a need to identify other home-based interventions that could provide similar mental health support.

Over the past two decades, there has been increasing research showing the effects of listening to music on mental health. A number of reviews have demonstrated the effects of regular music listening including in enhancing mental health in the general population,[17] reducing distress in premature infants[18] and reducing stress in adults.[19] Specifically in relation to depression, listening to music has been shown to reduce depression among adults with chronic pain,[20] psychiatric inpatients[21] and older adults.[17 22] In relation to well-being, music listening has been shown to be associated with better well-being not just in controlled interventions but also as a result of ordinary day-to-day listening. A Swedish study involving 500 older adults found associations between music listening and well-being, even when controlling for potential confounding variables.[23] Studies tracking daily activities have linked music listening with enhanced well-being both in the workplace and in the wider context of people's lives.[24 25] Further, music has also been shown to contribute to creating supportive healthy environments, connecting individuals with their emotions and promoting well-being.[26] Finally, theoretical studies have highlighted the role of music listening in enhancing affect, wellness and resources for recovery and quality of life.[27 28] Consequently, both directed music listening interventions and routine day-to-day music listening can affect levels of depression and well-being in a range of different populations.

Specifically in relation to the perinatal period, a few studies have suggested that music listening may be supportive for mental health. Listening to music for just 30 min has been found to reduce cortisol levels and anxiety in pregnant women, leading to recommendations that pregnant women might benefit from regular listening to music as a practice of relaxation (although the effects of regular listening were not tested).[29] A recent study found that women who listened to recorded music for 20 min a day for 12 weeks during their pregnancy had significant improvements in anxiety and depression.[30] However, the study did not track outcomes postnatally and involved a small sample of women. And a further study has found cross-sectional associations between listening to music and depression and well-being among new mothers, with more frequent listening associated with better mental health.[31] However, this study did not look longitudinally nor involved pregnant women.

Therefore, to date, despite promising results suggesting that listening to music can modulate mental health and well-being during the perinatal period, no studies have looked specifically at the impact of listening to music during pregnancy on depression and well-being post birth. In order to address this research gap, this study tracked a cohort of mothers across the perinatal period in order to ascertain whether there was a relationship between music listening during pregnancy and postnatal mental health.

## METHODS
### Participants and procedure
This study used data collected as part of a larger study exploring the impact of creative interventions on perinatal mental health. Women living in England in the last trimester of pregnancy (28 weeks or more) and the first 9 months postbirth (up to 40 weeks) were recruited from hospitals, general practices, mother and baby charities and through social media in England from October 2015 to March 2016, and completed an anonymous cross-sectional online questionnaire. Women in the final trimester

of pregnancy (28 weeks or more) were then invited to continue their participation in a longitudinal study. This involved providing another wave of data 3 months, 6 months and 9 months following the first date of data collection (which equated to providing baseline data T1 during pregnancy, T2 data in the first 3 months postbirth, T3 data in months 4–6 postbirth and T4 data in months 7–9 postbirth). From an initial sample of 550 mothers who consented to be involved in this longitudinal study, a total of 458 mothers provided T2 data (83%), 417 provided T3 data (75.8%) and 392 (71.3%) provided T4 data. The study received ethical approval and all participants gave informed consent prior to involvement in the research.

For this study, and in light of the literature review presented above, we hypothesised that listening during pregnancy would support well-being and reduce symptoms of PND in the first trimester postbirth. We therefore focused on women in the longitudinal study who had provided complete data on the variables we selected for analyses at both T1 and T2: 395 women. However, we also ran some exploratory follow-up analyses with women who had also provided complete data at T3 (n=299) in order to explore if effects were maintained.

### Patient public involvement

This study was developed as part of a wider grant that involved mothers, psychiatrists and health workers in the design of the research questions, the choice of measures and the recruitment for the study. We also involved these groups in the dissemination of the results.

### Measures

Symptoms of PND were measured using the EPDS, a 10-item scale used extensively both with pregnant women and new mothers, scored from 0 to 30 with 10+ indicative of possible symptoms of depression and higher scores of 13+ indicating more severe depression.[32]

Well-being was measured using the Short Warwick-Edinburgh Mental Well-being Scale, a scale that encompasses both hedonic and eudemonic well-being comprising seven items scored from 7 to 35 with higher scores representing higher levels of well-being. The raw scores were logit transformed prior to analysis.[33] The New Economics Foundation suggests five levels of well-being based on quintile analyses of data in the UK Understanding Society Survey, 2009: poor (<22), below average (22–24), average (25–26), good (27–28) and excellent (29– 35).

In addition, demographic variables assessed the women's number of weeks pregnant/postbirth, number of other children (0, 1, 2, 3 and 4+), household income (<£16 000, £16 000–£30 000, £31 000–£60 000, £61 000–£90 000, >£90 000), educational attainment (school to 16, sixth form/college, undergraduate degree, postgraduate degree), marital status (married vs not married), employment status (working vs not working), partner's employment status (working vs not working) and whether the woman had previously been diagnosed with either anxiety or depression.

Listening to music was categorised as 'rarely; a couple of times a week; every day <1 hour; every day 1–2 hours; every day 3–5 hours; every day 5+hrs'. While these analyses focused on quantity of music listening as a predictor, we also recorded genre of music listened to.

### Statistics

Data were analysed using Stata V.14. Multivariable linear regression models were used to explore the effects of listening to music on well-being and PND. Frequency of listening to music had a normal distribution, so was treated as a 6-point linear variable, with higher score indicating more frequent listening. For well-being and PND, we used raw scores. Model 1 was unadjusted, while model 2 adjusted for baseline well-being/depression, mother's age, maternal education status, household income and number of previous children, as well as how many weeks the baby was post birth, the mother's marital status at T2, whether she was working at T1 or T2, whether her partner was working at T2 and previous histories of both anxiety and depression.

All models displayed linearity as assessed by augmented partial residual plots with lowess smoothing; multicollinearity as assessed by checking variance inflation factors; normality as assessed using kernel density plots, standardised normal probability (P-P) plots and Q-Q plots; and there was no evidence of outliers or undue influence as assessed using added variable plots regressing each variable against all others, through stem and leaf plots, and through assessing covariance ratios, Cook's distance and leverage. The well-being regression models demonstrated homoscedasticity as assessed by plotting the residuals versus fitted (predicted) values and using the Breusch-Pagan test for heteroskedasticity. However, the depression regression models showed signs of heteroskedasticity, so robust standard errors were calculated.

Planned sensitivity analyses were then conducted in order to ascertain whether baseline mental health during pregnancy was a moderator of the association between listening to music and mental health postbirth. For this, we included an interaction term of Q1 mental health and Q1 listening habits in our regression models and then plotted two-way contour graphs to visualise the interaction (see online supplementary figure 1).

Although there were no significant demographic differences between those who did and did not provide data at Q3, we wanted to take account of potential minor demographic differences between those who provided data at Q3 and those who failed to. So the propensity score for non-response was calculated using the indicators listed in model 2 above (none of which significantly predicted missingness) and inverse probability weighting was applied to the T3 regression models. We confirmed goodness of fit using the Hosmer-Lemeshow test. Weighted analyses did not differ from unweighted analyses.

**Table 1** Demographic information on participants

|  | n=395 |
|---|---|
| Maternal age, μ (SD) | 31.9 (4.9) |
| Infant age, μ (SD) | 32.9 (4.1) |
| Marital status, % |  |
| Married | 69.3 |
| Cohabiting | 25.9 |
| In a relationship but living separately | 3.8 |
| Single | 1 |
| Partner working, % | 97.0 |
| Educational attainment, % |  |
| Education to 16) | 13.2 |
| Education to 18 | 16.5 |
| Undergraduate degree/qualification | 41.3 |
| Postgraduate degree/qualification | 29.1 |
| Household income, % |  |
| < £16 000 | 6.6 |
| £16 000–£30 000 | 11.1 |
| £31 000–£60 000 | 52.9 |
| £61 000–££90 000 | 17.5 |
| >£91 000 | 11.9 |
| Frequency of music listening, % |  |
| Rarely | 5.6 |
| A couple of times a week | 17.2 |
| Daily <1 hour | 34.4 |
| Daily 1–2 hours | 29.9 |
| Daily 3–5 hours | 8.6 |
| Daily 5+ hours | 4.3 |
| Genre of music listened to, % |  |
| Jazz | 21.0 |
| Pop | 93.7 |
| Rock | 57.7 |
| Classical | 34.2 |
| Folk | 22.8 |
| R&B | 42.8 |

## RESULTS
### Demographics
At T1, women had an average age of 31.9 years (SD=4.9, range 18–47) and an average of 32.9 weeks pregnant (SD=4.1, range 28–42). Further demographics are provided in table 1.

The average well-being score at T1 was 24.1 (SD=3.9, range 11.25–35), at T2 was 23.9 (SD=4.3, range 7–35) and at T3 was 23.8 (SD=4.1, range 7–35) (table 2). In order to calculate the change in well-being among these women, we analysed the difference in scores from T1 to T2 and T1 to T3. From T1 to T2, 39.2% of mothers experienced a decrease or at least one point in their well-being, while 30.4% experienced no change and 30.4% experienced an

**Table 2** Levels of well-being and postnatal depression during pregnancy (T1), 0–3 months postbirth (T2) and 4–6 months postbirth (T3)

|  | T1 | T2 | T3 |
|---|---|---|---|
| Well-being |  |  |  |
| Poor (<22) | 29.4% | 31.1% | 31.6% |
| Below average (22–24) | 25.6% | 24.1% | 25.4% |
| Average (25–26) | 23.5% | 21.3% | 20.9% |
| Good (27–28) | 10.1% | 11.7% | 13.7% |
| Excellent (29–35) | 11.4% | 11.9% | 8.5% |
| Depression |  |  |  |
| EPDS <10 | 74.7% | 72.2% | 73.3% |
| EPDS >=10 | 25.3% | 27.9% | 26.7% |

EPDS, Edinburgh Postnatal Depression Scale.

improvement of at least one point. From T1 to T3, 41.4% of mothers experienced a decrease or at least one point in their well-being, while 25.7% experienced no change and 32.9% experienced an improvement of at least one point.

As with well-being, we calculated the change in symptoms of PND from T1 to T2 and T1 to T3 (table 2). From T1 to T2, 43.6% of mothers experienced an increase in the number of symptoms of PND they were experiencing, while 11.2% experienced no change and 45.3% of mothers experienced an improvement in symptoms. From T1 to T3, 56.4% of mothers experienced an increase in the number of symptoms of PND they were experiencing, while 11.6% experienced no change and 43.6% experienced an improvement in symptoms. In terms of the interaction between well-being and symptoms of PND, there was a large correlation between the two at T1 ($r=-0.67$, $p<0.001$), T2 ($r=-0.76$, $p<0.001$) and T3 ($r=-0.77$, $p<0.001$), suggesting 45%-59% shared variance.

### Regression results
Listening to music while pregnant was associated with higher raw well-being scores 0–3 months postbirth, even when accounting for potential confounding variables, with greater frequency associated with greater effects (B=0.40, SE=0.15, 95% CI 0.10 to 0.70) (see table 3). However, our exploratory analyses showed that effects were not evident 4–6 months postbirth. There was also an association between listening to music while pregnant and raw scores of symptoms of PND, even when accounting for potential confounding variables, with more frequent listening to music during pregnancy associated with lower symptoms of PND in the first 3 months postbirth (B=−0.39, SE=0.19, 95% CI −0.76 to −0.03). As with well-being, these results were no longer evident by months 4–6 postbirth.

### Further analyses
Sensitivity analyses of the well-being regression models explored the potential moderating effect of mental health during pregnancy. Our analyses showed there was

**Table 3** Associations between listening to music during pregnancy on well-being and symptoms of postnatal depression postbirth

| | Well-being | | | | Symptoms of PND | | | |
|---|---|---|---|---|---|---|---|---|
| | B | SE | 95% CI | P value | B | SE | 95% CI | P value |
| Months 0–3 postbirth (n=395) | | | | | | | | |
| Model 1 | 0.63 | 0.19 | 0.26 to 1.00 | 0.001 | −0.52 | 0.23 | −0.98 to −0.06 | 0.028 |
| | $R^2$=0.03, F(1, 393)=11.44, p=0.001 | | | | $R^2$=0.01, F(1,393)=4.74, p=0.03 | | | |
| Model 2 | 0.40 | 0.15 | 0.10 to 0.70 | 0.01 | −0.39 | 0.19 | −0.76 to −0.03 | 0.036 |
| | $R^2$=0.43, F(18,376)=15.97, p<0.001 | | | | $R^2$=0.39, F(18,376)=9.58, p<0.001 | | | |
| Months 3–6 postbirth (n=299) | | | | | | | | |
| Model 1 | 0.33 | 0.21 | −0.079 to 0.74 | 0.11 | −0.11 | 0.24 | −0.59 to 0.36 | 0.63 |
| | $R^2$=0.01, F(1,305)=2.17, p=0.14 | | | | $R^2$=0.003, F(1,301)=0.86, p=0.35 | | | |
| Model 2 | 0.12 | 0.16 | −0.20 to 0.43 | 0.47 | −0.02 | 0.20 | −0.41 to 0.36 | 0.90 |
| | $R^2$=0.44, F(18,284)=11.22, p<0.001 | | | | $R^2$=0.37, F(18,280)=7.19, p<0.001 | | | |

Model 1: unadjusted; model 2: adjusted for baseline well-being/depression, mother's age, maternal education status, household income and number of previous children, as well as how many weeks the baby was postbirth, and the mother's marital status at T2, whether she was working at T1 or T2 and whether her partner was working at T2 and previous histories of both anxiety and depression.
*P<0.05.
**P<0.01.
***P<0.001.
PND, postnatal depression.

a significant interaction between listening and baseline well-being (B=−0.10 SE=0.04, 95% CI −0.17 to −0.02), with a two-way contour graph showing that listening to music particularly seemed to support those with lower well-being at baseline (see online supplementary figure 1A). There was also a significant interaction between listening and baseline depression (B=−0.09, SE=0.04, 95% CI −0.17 to −0.01), with a two-way contour graph showing that listening to music particularly seemed to support those who were showing symptoms of PND (evidenced with a score of 10+ at baseline) (see online supplementary figure 1B).

Finally, in order to try and ascertain whether listening to music led to changes in mental health or whether mental health led to changes in listening habits, we ran additional analyses reversing the variable order. While this does not confirm potential causal mechanisms, it can give an indication as to whether mental health can predict listening behaviours and therefore support hypotheses about temporal precedence. There was no evidence that levels of well-being during pregnancy were associated with the likelihood of listening to music either 3 or 6 months postbirth. However, there was some indication that depression symptoms in the final trimester of pregnancy were associated with listening habits 3 months postbirth (β=−0.03, SE=0.01, 95% CI −0.05 to −0.01, p=0.003).

## DISCUSSION

This study explored associations between listening to music in the final trimester of pregnancy and mental health and well-being in mothers postbirth. Listening was found to be associated with higher levels of well-being and reduced symptoms of PND in the first 3 months postbirth, even when adjusting for baseline mental health and potential confounding variables. These results appear to be particularly found among women with lower levels of well-being at baseline. These findings echo the few existing studies in showing that listening to music is associated with better mental health in the perinatal period.[30 31] However, to the authors' knowledge, this is the first study to show that listening to music during pregnancy is longitudinally associated with better mental health postbirth.

Across both symptoms of PND and well-being, however, associations were only found for the first 3 months postbirth, and had disappeared by the second quartile postbirth. The hypnosis study previously described also found results within the first 3 months postpartum (weeks 2 and 10) but did not measure beyond this, so there is little data available against which to benchmark these findings.[16] Nevertheless, the immediate period postbirth has been highlighted as being of particular challenge for new mothers, with the transition into assuming maternal tasks and adjusting to the new role lasting until around the third month postpartum.[13 34] It is possible, therefore,

that any effects of music listening during the prenatal period are of most value during this transition period, but become less significant once mothers and their babies become more settled.

A key question is how listening to music is associated with better mental health and well-being in the postnatal period. There are a number of potential explanations. First, studies involving psychological tests, neuroimaging, biomarker analyses and ethnographic observations have shown that listening to music can have marked effects on stress and anxiety.[25] Specifically in relation to pregnant women, listening to music for just 30 min can reduce cortisol levels and anxiety.[29] Wider studies involving listening to music have shown it to be particularly effective at reducing psychological and physiological responses to stress, especially when people deliberately listen to music in order to help them relax.[35 36] This effect of music on stress has in turn been linked specifically through to theories around well-being,[25 37] with a wide literature linking stress and anxiety with both mental health and well-being.[38 39] It is proposed that high levels of anxiety might hinder women's adaptation to motherhood in the initial postpartum period, with negative effects on well-being.[40] Consequently, it is possible that the relaxing effects of listening to music during the pregnancy period help to act as a buffer for feelings of stress and anxiety, thereby supporting mothers in maintaining their adaptation and leading to enhanced well-being.

Another potential explanation relates to the effects of music on mood. Mood regulation has been identified as one of the prime reasons why people listen to music, with models of mood regulation by music highlighting its effects on mood-related subjective experience (including the intensity and clarity of moods), physiological responses (such as energy levels and movement) and behaviours (such as their ability to express emotions).[41] Music listening has been found to modulate depression and well-being.[42 43] Early low mood during the prenatal period is directly associated with lower well-being and postnatal depression postbirth,[44] leading to propositions that interventions that deliberately attempt to cultivate positive emotions, such as relaxation therapies and interventions focused on finding positive meaning, could directly optimise health and well-being in this population. Consequently, it is possible that another route by which listening to music in the third trimester of pregnancy is associated with improvements in mental health and well-being is via enhancing mood.

Finally, a third explanation is that listening to music in itself did not have an effect postbirth but did enhance coping skills in women while they were still pregnant, which in turn led to higher well-being postbirth. Music listening has been linked with both problem-oriented coping and emotion-oriented coping, specifically with results showing that problem-oriented coping by music listening in women is linked to lower depression levels.[45] Life transitions (such as the perinatal period) depend on both health and well-being and also on appraisal and coping responses. In the hypnosis study previously mentioned, the authors proposed that the intervention during pregnancy helped mothers to maintain and enhance their well-being while pregnant, which in turn influenced their appraisal of the perinatal transition period and supported their coping responses.[16] It is possible that a similar process took place through listening to music, with listening to music supporting coping in the prenatal period, which encouraged mothers' own coping skills, which in turn led to better coping during and postbirth and consequent higher well-being. Indeed, even just in relation to the birth, a number of birth preparation courses focus on relaxation and mood optimisation, which have been shown to lead to less negative affect and better coping during labour and delivery.[46] Given that a significant predictor of PND is the birth experience, enhanced coping prior to the birth, perhaps through music listening, could be an important factor in postnatal well-being.[47]

This study has a number of limitations. First, the study followed a cohort of women rather than being interventional, so it is not possible to confirm causality. However, the study had a longitudinal design, there was no evidence of reverse causality in relation to well-being, there are plausible proposed mechanistic explanations and there is a strong body of previous literature causally linking music and mental health in other populations. So this study provides promising preliminary evidence that remains to be tested in a future experimental design. A second limitation is that the population in this study was not nationally representative. Nevertheless, there was a clear spread of participants from varying socioeconomic backgrounds as well as variations in the levels of exposure and outcome variables. So the data set provides interesting and suitable preliminary data on the longitudinal associations between music listening and mental health. Third, this study explored the impact of all music listening, not specifying particular genres. Previous research has suggested that certain genres of music (or more specifically compositional aspects of music such as its valence and arousal levels) can lead to different responses, such as variations in relaxation or mood.[48] However, most of these genre-specific effects have been found in tightly controlled laboratory-based studies, and literature from real-world interaction with music has suggested that musical preference might be more important in determining the effects of music.[49] This study followed these real-world studies in recording what genres people did listen to but measuring the quantity of listening based on preference rather than genre. Future studies could explore the impact of different genres on mental health in the perinatal period.

In conclusion, this study provides the first preliminary evidence that listening to music during the third trimester of pregnancy could be protective against symptoms of PND and low well-being in the first 3 months postbirth. Music listening is an attractive intervention in that it is readily available to people from all echelons of society regardless of socioeconomic status, educational attainment or cultural background. It can be carried out in a range of contexts so is not restricted to particular places or times. It is also inexpensive: indeed the majority of

women in the Western countries have access to recorded music already. Finally, there are no obvious side effects from listening to music. Consequently, listening to music could be recommended as a way of supporting pregnant women, in particular those who demonstrate low well-being or symptoms of PND.

**Acknowledgements**  The study team acknowledge the support of the National Institute of Health Research Clinical Research Network (NIHR CRN). The authors would like to thank the hospitals involved as Participant Identification Centres as well as Miss Sunita Sharma, Prof Aaron Williamon and Sarah Yorke for their support with the study.

**Contributors**  DF and RP designed the study and collected data. DF ran the analyses and drafted the paper. Both authors critically revised the manuscript and approved it for submission.

**Funding**  The study was funded by Arts Council England Research Grants Fund, grant number 29230014 (Lottery). DF is supported by the Wellcome Trust [205407/Z/16/Z].

**Competing interests**  None declared.

**Patient consent**  Not required.

**Ethics approval**  UK NHS Research Ethics Committee.

**Provenance and peer review**  Not commissioned; externally peer reviewed.

**Data sharing statement**  The data are available from the authors upon request.

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
