## [Reviewer comments · BMJ Open]

ARTICLE DETAILS

TITLE (PROVISIONAL)	Could listening to music during pregnancy be protective against postnatal depression and poor wellbeing post-birth? Longitudinal associations from a preliminary prospective cohort study.
AUTHORS	Fancourt, Daisy; Perkins, Rosie

VERSION 1 – REVIEW

REVIEWER	dr Łucja Bieleninik Institute of Psychology, University of Gdańsk, Gdańsk, Poland
REVIEW RETURNED	28-Jan-2018

GENERAL COMMENTS	A manuscript entitled "Could listening to music during pregnancy be protective against postnatal depression and poor wellbeing post-birth? Longitudinal associations from a preliminary prospective cohort study" is well aligned with the BMJ Open mission statement. Authors undertook an unique and interesting research to the field of music, expanding listening to music to medical health care service. Overall, this project offers a good cross-disciplinary ground and findings have clinical applicability to the field of obstetrics, psychiatry as well as clinical psychology. A topic of the manuscript is important. Stress and anxiety are common in pregnancy, especially pregnancy-related complications; and shown to have adverse effects on both maternal and infant health outcomes in postnatal period. Moreover, pregnant women and their babies share hormones, so there is a close connection between the emotional well-being of the mother and that of the child she carries. Thus, music which helps the pregnant women to relax seems to be a good non-pharmacological intervention in reducing the level of anxiety and stress. Mothers' psychological well-being and postpartum depression are the most important factors affecting mother-infant relationship in the postnatal period, which plays a central role in the child's socio-emotional development and formation of secure attachment. That is why decreasing maternal postnatal depression and increasing well-being post-birth are crucial aims for positively impacting long-term outcomes of both mother and their babies. A favorable effect of music during pregnancy was confirmed in some research; however, several areas require further investigation. This study tracked a cohort of 395 mothers across the perinatal period in order to explore associations between listening to music in the third trimester of pregnancy and mental health and wellbeing at 3 and 6-months post-birth. A prospective cohort study followed the STROBE guidelines, agreed core outcomes and standardized reporting measures was developed and implemented. The manuscript is characterized by appropriate study design, methods,
---

	statistical analysis as well as quality of results presentation, conclusions and discussion. Limitations are properly acknowledged as well. Results have shown that listening during pregnancy is associated with higher level of wellbeing and lower level of postnatal depression in the first 3 months post-birth. Authors analyzed the quantity of music listening as a predictor. Mothers listened the type of music in accordance with their preferences. The choice of music is an important component, with stress reduction being dependent on the music preference of the participant. To my knowledge, there is a limited number of studies that explored whether listening to music during pregnancy is longitudinally associated with mother's psychological well-being and postnatal depression. Therefore, this study fills a gap in knowledge by assessing the longitudinal correlation between listening to music during pregnancy and pregnant woman's outcomes postnatally. However, this project was designed as a cohort study (no interventional study), therefore there is no possible to calculate an effectiveness of receptive music on mother's outcomes. Proposed project provides promising preliminary evidence that remains to be tested in a future by rigorously designed and adequately powered study using standardized outcome measures and clearly articulated intervention. In addition, for the next project it's recommended to take into consideration the type of pregnancy and type of delivery as predictors in order to ascertain whether listening to music could be particular value for women with or without history of pregnancy-related complications as well as with or without history of preterm birth.
--	---

REVIEWER	Charles Opondo
	London School of Hygiene & Tropical Medicine, UK
REVIEW RETURNED	09-Feb-2018

GENERAL COMMENTS	This investigates the effect of music in modulating the mental health and well-being of pregnant women and up to nine months after delivery. Given the increasing prevalence of poor mental health and well-being in women during this period, and the severe (and often long-term) consequences for their children and families, this is an important study, and the authors have done a commendable job in conducting and describing it. There are nevertheless a number of issues which I would recommend that the authors address in their revisions to the manuscript: 1. In the abstract, the authors should revise the 'primary and secondary outcome measures' section to be more precise. For example, the first line refers to listening to music, which was the main exposure and not an outcome. They also refer to multivariate linear regression analyses (i.e. regression of multiple outcomes in a single model), but it appears that they in fact conducted multivariable regression models (a single outcome but potentially multiple explanatory variables (see https://www.ncbi.nlm.nih.gov/pmc/articles/PMC3518362/ for more on this). There are further clarifications on the reporting of results, details of which I'll address below (e.g. how much listening is associated with higher levels of wellbeing, in the 'results' section of the abstract, as at the moment it reads as if 'listening' was a binary variable).
--

2. In the introduction, the authors have provided an extensive background of the study and other relevant studies. This was mostly fine, however a lot of the previous evidence could have been cited in more useful ways. For example, in the first paragraph, the authors report the association between depression and positive experience of motherhood by citing a correlation coefficient; while correlation and association are related concepts, they are not the same and should not be reported interchangeably in this way. Similarly, in the fourth paragraph, the authors refer to correlations between listening to music and wellbeing when the context appears to suggest that they are in fact referring to association. In the fifth paragraph, I thought that it would have been more helpful to the reader to report some metrics where it was reported that listening to music led to improvements in anxiety and depression (i.e. what level of listening, how much, and how much improvement in the outcome and on what scale/measure). Without this, the reader has to check than information from the reference which should not be necessary, at least not for such key results

3. In the methods, in the second paragraph, the authors have presented some results - these should be moved to the results section.

However, and more importantly, there needs to be a much clearer description of the analysis. It is well and good that approaches to regression diagnostics have been described. However, the regression models themselves have not been clearly described. I could not tell whether the outcomes were raw scores, ordered grouped scores, or change-from-baseline scores.

Additionally, what are described as 'planned sensitivity analyses' seem to me to be subgroup analyses which were likely conducted by fitting interaction terms in the regression models - again this is not clearly articulated (and I could even be wrong about what the authors actually did, although according to the description this would normally be the approach to the analysis).

Given that this was a longitudinal study, it was an inefficient use of data to limit the analysis to complete cases at T1 and T2, and then to only use T3 data for exploratory analyses. A single hierarchical longitudinal analysis including all data from T1 to T4 would be most efficient, as it would use data from all cases, including observations with missing data on some covariates, would suitably adjust for baseline, and would be statistically most powerful.

4. Results: the first set of results tables should have included a description of the participants on all key characteristics, not just the outcome measures only.

The results presenting correlations between measures at separate time points are not useful at all and should be removed - measurements from the same individuals over time are naturally expected to be correlated with each other, and a statistically significant correlation between such measures is not at all an insightful finding. Furthermore, it was not mentioned at all in the statistical methods section that this kind of analysis would be performed (the results should reflect the methods).

The regression results are hard to make sense of because the explanatory and outcome variables have not been clearly described.

	For example, it is hard to tell what quantity of listening is associated with higher well-being, and also not possible to tell exactly what 'higher well-being levels' means. The same applies to PND. The current description appears to imply that listening to music is a binary variable, yet previous statements suggest that it is not. Under 'further analyses' it appears that tests for interaction have been conducted, however, there is no mention of this in the description of methods. And if this is the case, then it is the p-values for the interaction, rather than the individual effect estimate p-values, that assesses the evidence for a difference in effect across the subgroups assessed. Lastly, the assessment of what the potential causal mechanism or order of occurrence of potential causes and effects in the last paragraph of the results is, in my opinion, incorrect. Simply reversing the 'causes' and 'effects' in a regression model does not provide any evidence about what the likely causal mechanism is. Indeed, in many cases, the evidence of association is similar even when this is done.
--	---

REVIEWER	Gemma Hammerton University of Bristol
REVIEW RETURNED	07-Mar-2018

GENERAL COMMENTS	This is a statistical review for a paper that examines whether listening to music during pregnancy is associated with lower symptoms of depression and higher well-being in mothers post-birth. The paper is very clear and well written and the analytical methods are appropriate and well described; however, I have a few minor comments and suggestions for improvement, outlined below by section. More information could be provided on those with missing data at T2 and T3 and how the IPW was performed. Specifically, did the sample that provided data at T1 and T2 (395 women) differ from the original sample (550 women)? If so why was IPW not used for these analyses. Which variables were associated with nonresponse at T3 and which indicators were used in creating the propensity score? Was the Hosmer-Lemeshow test used to assess the fit of the missingness model? Did the weighted analyses differ from the unweighted? It is not clear whether baseline measures of depression/ well-being were hypothesised to be confounders or moderators of the association between listening to music and later depression/ well-being. These measures were treated as confounders in the main analyses, but moderators in the sensitivity analyses. If these baseline measures are considered to be potential moderators, then an interaction should be tested between baseline depression/ wellbeing and listening to music rather than simply performing subgroup analyses and reporting a significant association in one group but not in the other. Associations within each subgroup are in the same direction and do not necessarily statistically differ from each other. Lastly, I wondered whether the authors considered the use of a latent growth curve (or multi-level model) to examine change in symptoms over time (perhaps also including T4). This would hold a number of advantages including using FIML estimation to
--

	incorporate those with missing data, and addressing measurement error in the repeated measures. It would also be interesting to examine whether listening to music was associated with a change in depression/ well-being over time accounting for initial levels at baseline.
--	--

VERSION 1 – AUTHOR RESPONSE

Response to reviewers

Reviewer: 1

Reviewer Name: dr Łucja Bieleninik

Institution and Country: Institute of Psychology, University of Gdańsk, Gdańsk, Poland

A manuscript entitled "Could listening to music during pregnancy be protective against postnatal depression and poor wellbeing post-birth? Longitudinal associations from a preliminary prospective cohort study" is well aligned with the BMJ Open mission statement. Authors undertook an unique and interesting research to the field of music, expanding listening to music to medical health care service. Overall, this project offers a good cross-disciplinary ground and findings have clinical applicability to the field of obstetrics, psychiatry as well as clinical psychology.

A topic of the manuscript is important. Stress and anxiety are common in pregnancy, especially pregnancy-related complications; and shown to have adverse effects on both maternal and infant health outcomes in postnatal period. Moreover, pregnant women and their babies share hormones, so there is a close connection between the emotional well-being of the mother and that of the child she carries. Thus, music which helps the pregnant women to relax seems to be a good non-pharmacological intervention in reducing the level of anxiety and stress. Mothers' psychological well-being and postpartum depression are the most important factors affecting mother-infant relationship in the postnatal period, which plays a central role in the child's socio-emotional development and formation of secure attachment. That is why decreasing maternal postnatal depression and increasing well-being post-birth are crucial aims for positively impacting long-term outcomes of both mother and their babies. A favorable effect of music during pregnancy was confirmed in some research; however, several areas require further investigation.

This study tracked a cohort of 395 mothers across the perinatal period in order to explore associations between listening to music in the third trimester of pregnancy and mental health and wellbeing at 3 and 6-months post-birth. A prospective cohort study followed the STROBE guidelines, agreed core outcomes and standardized reporting measures was developed and implemented. The manuscript is characterized by appropriate study design, methods, statistical analysis as well as quality of results presentation, conclusions and discussion. Limitations are properly acknowledged as well. Results have shown that listening during pregnancy is associated with higher level of wellbeing and lower level of postnatal depression in the first 3 months post-birth. Authors analyzed the quantity of music listening as a predictor. Mothers listened the type of music in accordance with their preferences. The choice of music is an important component, with stress reduction being dependent on the music preference of the participant.

To my knowledge, there is a limited number of studies that explored whether listening to music during pregnancy is longitudinally associated with mother's psychological well-being and postnatal depression. Therefore, this study fills a gap in knowledge by assessing the longitudinal correlation between listening to music during pregnancy and pregnant woman's outcomes postnatally. However, this project was designed as a cohort study (no interventional study), therefore there is no possible to calculate an effectiveness of receptive music on mother's outcomes. Proposed project provides promising preliminary evidence that remains to be tested in a future by rigorously designed and adequately powered study using standardized outcome measures and clearly articulated intervention. In addition, for the next project it's recommended to take into consideration the type of pregnancy and type of delivery as predictors in order to ascertain whether listening to music could be particular value for women with or without history of pregnancy-related complications as well as with or without history of preterm birth.

We'd like to thank Dr Bieleninik for these comments. We're pleased that you found the study's design, methods, analyses and presentation appropriate and the topic an important one. We agree with your recommendations for future research studies and hope that this article will help to support the development of such future work.

Reviewer: 2

Reviewer Name: Charles Opondo

Institution and Country: London School of Hygiene & Tropical Medicine, UK

This investigates the effect of music in modulating the mental health and well-being of pregnant women and up to nine months after delivery. Given the increasing prevalence of poor mental health and well-being in women during this period, and the severe (and often long-term) consequences for their children and families, this is an important study, and the authors have done a commendable job in conducting and describing it.

We'd like to thank Dr Opondo for his comments and are pleased that he commends the manuscript. Many thanks also for the detailed feedback that follows.

There are nevertheless a number of issues which I would recommend that the authors address in their revisions to the manuscript:

1. In the abstract, the authors should revise the 'primary and secondary outcome measures' section to be more precise. For example, the first line refers to listening to music, which was the main exposure and not an outcome. They also refer to multivariate linear regression analyses (i.e. regression of multiple outcomes in a single model), but it appears that they in fact conducted multivariable regression models (a single outcome but potentially multiple explanatory variables (see <https://www.ncbi.nlm.nih.gov/pmc/articles/PMC3518362/> for more on this). There are further clarifications on the reporting of results, details of which I'll address below (e.g. how much listening is associated with higher levels of wellbeing, in the 'results' section of the abstract, as at the moment it reads as if 'listening' was a binary variable).

We have now updated the abstract to make it clearer that listening to music was the exposure and that depression and wellbeing were the outcomes. We are also grateful to Dr Opondo for noticing the mistake regarding the regression description: we have corrected multivariate to multivariable.

2. In the introduction, the authors have provided an extensive background of the study and other relevant studies. This was mostly fine, however a lot of the previous evidence could have been cited in more useful ways. For example, in the first paragraph, the authors report the association between depression and positive experience of motherhood by citing a correlation coefficient; while correlation and association are related concepts, they are not the same and should not be reported interchangeably in this way. Similarly, in the fourth paragraph, the authors refer to correlations between listening to music and wellbeing when the context appears to suggest that they are in fact referring to association. In the fifth paragraph, I thought that it would have been more helpful to the reader to report some metrics where it was reported that listening to music led to improvements in anxiety and depression (i.e. what level of listening, how much, and how much improvement in the outcome and on what scale/measure). Without this, the reader has to check than information from the reference which should not be necessary, at least not for such key results

We have now clarified our use of the terms correlation and association in the introduction. For the first example, this was in fact a simple correlation. We have corrected the second point you note to describe it as an association. We have provided a few more details about the previous studies, particularly pertaining to the amount of listening, but as the authors of these previous studies did not quantify the amount of improvement beyond regression coefficients we would prefer to refer readers to the articles themselves for any further information.

3. In the methods, in the second paragraph, the authors have presented some results - these should be moved to the results section.

We have now moved the demographic description of participants to the results section.

However, and more importantly, there needs to be a much clearer description of the analysis. It is well and good that approaches to regression diagnostics have been described. However, the regression models themselves have not been clearly described. I could not tell whether the outcomes were raw scores, ordered grouped scores, or change-from-baseline scores.

We have now provided this information clarifying that it was raw scores that we were using and giving more detail about the treatment of the independent variable (see comment below).

Additionally, what are described as 'planned sensitivity analyses' seem to me to be subgroup analyses which were likely conducted by fitting interaction terms in the regression models - again this is not clearly articulated (and I could even be wrong about what the authors actually did, although according to the description this would normally be the approach to the analysis).

We apologise for the lack of clarity here. In line with the third reviewer's comments, we have now updated the methods and further analyses. Instead of stratifying by baseline mental health as we did before, we now report the results of moderation analyses (fitting interaction terms as you say), showing the coefficient and p value for interaction and then providing contour graphs for clarity.

Given that this was a longitudinal study, it was an inefficient use of data to limit the analysis to complete cases at T1 and T2, and then to only use T3 data for exploratory analyses. A single hierarchical longitudinal analysis including all data from T1 to T4 would be most efficient, as it would use data from all cases, including observations with missing data on some covariates, would suitably adjust for baseline, and would be statistically most powerful.

We appreciate your suggestion about a hierarchical longitudinal analysis being potentially more efficient here. However, our hypothesis was specifically related to listening during pregnancy and effects in the first trimester post-birth. Given that (as our literature search shows) research has generally previously focused just on the first trimester post birth, we did not initially hypothesise that listening during pregnancy would affect results 6 months later and initially only planned to analyse T1 and T2. We decided, following our initial analyses, to carry out an exploratory test of T3 to see whether results seemed to hold longer, which is why this T1-T3 analysis is described as 'exploratory'. But as this was not our initial aim and as we did not design the study with this additional analysis in mind, we propose retaining our current analyses, which focus on the primary hypothesis.

4. Results: the first set of results tables should have included a description of the participants on all key characteristics, not just the outcome measures only.

This description is now provided at the start of the results section under the heading 'demographics'.

The results presenting correlations between measures at separate time points are not useful at all and should be removed - measurements from the same individuals over time are naturally expected to be correlated with each other, and a statistically significant correlation between such measures is not at all an insightful finding. Furthermore, it was not mentioned at all in the statistical methods section that this kind of analysis would be performed (the results should reflect the methods).

We have now removed these correlations.

The regression results are hard to make sense of because the explanatory and outcome variables have not been clearly described. For example, it is hard to tell what quantity of listening is associated with higher well-being, and also not possible to tell exactly what 'higher well-being levels' means. The same applies to PND. The current description appears to imply that listening to music is a binary variable, yet previous statements suggest that it is not.

Following our comments above, we have now provided more information on the explanatory and outcome variables.

Under 'further analyses' it appears that tests for interaction have been conducted, however, there is no mention of this in the description of methods. And if this is the case, then it is the p-values for the interaction, rather than the individual effect estimate p-values, that assesses the evidence for a difference in effect across the subgroups assessed.

Please see our comment about our change in approach regarding the planned sensitivity analyses above.

Lastly, the assessment of what the potential causal mechanism or order of occurrence of potential causes and effects in the last paragraph of the results is, in my opinion, incorrect. Simply reversing the 'causes' and 'effects' in a regression model does not provide any evidence about what the likely causal mechanism is. Indeed, in many cases, the evidence of association is similar even when this is done.

We agree that reversing cause and effect does not provide evidence about likely causal mechanisms, but it does give us an indication of temporal precedence through showing whether mental health can predict listening habits, and it is a technique widely reported in other studies. We have now caveated our explanation to explain that we are not intending to judge causal mechanisms with this approach. We feel it is noteworthy that our results were only found in one direction rather than the evidence of association being similar both ways.

Reviewer: 3

Reviewer Name: Gemma Hammerton

Institution and Country: University of Bristol

This is a statistical review for a paper that examines whether listening to music during pregnancy is associated with lower symptoms of depression and higher well-being in mothers post-birth. The paper is very clear and well written and the analytical methods are appropriate and well described; however, I have a few minor comments and suggestions for improvement, outlined below by section.

We'd like to thank Dr Hammerton for her comments and are pleased that she feels the statistical methods are appropriate and well described. Thank you also for the feedback that follows.

More information could be provided on those with missing data at T2 and T3 and how the IPW was performed. Specifically, did the sample that provided data at T1 and T2 (395 women) differ from the original sample (550 women)? If so why was IPW not used for these analyses. Which variables were associated with nonresponse at T3 and which indicators were used in creating the propensity score? Was the Hosmer-Lemeshow test used to assess the fit of the missingness model? Did the weighted analyses differ from the unweighted?

There were no significant demographic differences between those who did and did not provide data which is why we did not weight all the analyses. But following your suggestion, we have now included more details as to which indicators were used in creating the propensity score for the T3 analyses where we had missing data. Yes the Hosmer-Lemeshow test was used and is now reported. The weighted analyses did not differ from unweighted analyses and this is also now clarified.

It is not clear whether baseline measures of depression/ well-being were hypothesised to be confounders or moderators of the association between listening to music and later depression/ well-being. These measures were treated as confounders in the main analyses, but moderators in the sensitivity analyses. If these baseline measures are considered to be potential moderators, then an interaction should be tested between baseline depression/ wellbeing and listening to music rather than simply performing subgroup analyses and reporting a significant association in one group but not in the other. Associations within each subgroup are in the same direction and do not necessarily statistically differ from each other.

Thank you for this comment. We have followed your advice and changed our approach from stratification to assessing the interaction and we have also provided contour graphs so the reader can see the effects of this interaction.

Lastly, I wondered whether the authors considered the use of a latent growth curve (or multi-level model) to examine change in symptoms over time (perhaps also including T4). This would hold a number of advantages including using FIML estimation to incorporate those with missing data, and addressing measurement error in the repeated measures. It would also be interesting to examine whether listening to music was associated with a change in depression/ well-being over time accounting for initial levels at baseline.

We appreciate this suggestion and refer back to our response to Dr Opondo. We only initially intended to explore T1 and T2, but added our exploratory analysis of T3 to provide supplementary detail that could support future studies. We would prefer to leave such analyses for a future study.

VERSION 2 – REVIEW

REVIEWER	Gemma Hammerton University of Bristol, United Kindom
REVIEW RETURNED	31-Mar-2018

GENERAL COMMENTS	The authors have addressed all of my previous comments, and I only have one remaining minor comment for clarification. It would be useful if the authors could add a brief foot note to Supplementary Figure 1 to explain how to interpret a contour graph for those not familiar.
--

REVIEWER	Charles Opondo London School of Hygiene & Tropical Medicine
REVIEW RETURNED	06-Apr-2018

GENERAL COMMENTS	The authors have responded appropriately to my and other reviewers' comments, and I'm happy to recommend that the manuscript be accepted for publication pending a few additional minor revisions:  1. the description of participants would be easier to understand if included in table 1 (which is the typical way of describing the sample) rather than presenting a textual description as has been done. This should include the group counts and proportions/percentages. Please see https://www.ncbi.nlm.nih.gov/pmc/articles/PMC4008059/ for some examples 2. in the table of results (Table 2), I would suggest that the authors indicate the number of observations in Models 1 and 2; these provide additional useful descriptions of the models and will be useful to future systematic reviews and meta-analyses. I would also suggest the reporting of actual p-values, which are more informative than simply indicating $p < 0.05/0.01/0.001$. If the authors need to replace one of the existing columns then I'd suggest getting rid of the t-statistic (which is redundant once B and SE are presented, given that t is approximately equal to B/SE). Please see https://www.ncbi.nlm.nih.gov/pmc/articles/PMC5101968/ for some examples
--

VERSION 2 – AUTHOR RESPONSE

Reviewer: 3

Reviewer Name: Gemma Hammerton

Institution and Country: University of Bristol, United Kindom

Please state any competing interests or state 'None declared': None declared

Please leave your comments for the authors below The authors have addressed all of my previous comments, and I only have one remaining minor comment for clarification. It would be useful if the

authors could add a brief foot note to Supplementary Figure 1 to explain how to interpret a contour graph for those not familiar.

We are pleased that Reviewer 3 is now happy with the manuscript and have made the final change you suggest, adding a footnote explaining the interpretation of the contour graph.

Reviewer: 2

Reviewer Name: Charles Opondo

Institution and Country: London School of Hygiene & Tropical Medicine

Please state any competing interests or state 'None declared': None declared

Please leave your comments for the authors below The authors have responded appropriately to my and other reviewers' comments, and I'm happy to recommend that the manuscript be accepted for publication pending a few additional minor revisions:

We are pleased that Reviewer 2 is now happy with the manuscript. We have made the minor changes proposed below in the manuscript.

1. the description of participants would be easier to understand if included in table 1 (which is the typical way of describing the sample) rather than presenting a textual description as has been done. This should include the group counts and proportions/percentages. Please see <https://www.ncbi.nlm.nih.gov/pmc/articles/PMC4008059/> for some examples

We have now taken out the text and replaced it with a table as suggested.

2. in the table of results (Table 2), I would suggest that the authors indicate the number of observations in Models 1 and 2; these provide additional useful descriptions of the models and will be useful to future systematic reviews and meta-analyses. I would also suggest the reporting of actual p-values, which are more informative than simply indicating $p < 0.05/0.01/0.001$. If the authors need to replace one of the existing columns then I'd suggest getting rid of the t-statistic (which is redundant once B and SE are presented, given that t is approximately equal to B/SE). Please see <https://www.ncbi.nlm.nih.gov/pmc/articles/PMC5101968/> for some examples

We have now done so.